# Quorum Sensing Inhibition by Marine Bacteria

**DOI:** 10.3390/md17070427

**Published:** 2019-07-23

**Authors:** Anabela Borges, Manuel Simões

**Affiliations:** LEPABE, Department of Chemical Engineering, Faculty of Engineering, University of Porto, Rua Dr. Roberto Frias, s/n, 4200-465 Porto, Portugal

**Keywords:** antimicrobial resistance, antipathogenic and antivirulence, quorum sensing, quorum quenching, marine bacteria, selective pressure

## Abstract

Antibiotic resistance has been increasingly reported for a wide variety of bacteria of clinical significance. This widespread problem constitutes one of the greatest challenges of the twenty-first century. Faced with this issue, clinicians and researchers have been persuaded to design novel strategies in order to try to control pathogenic bacteria. Therefore, the discovery and elucidation of the mechanisms underlying bacterial pathogenesis and intercellular communication have opened new perspectives for the development of alternative approaches. Antipathogenic and/or antivirulence therapies based on the interruption of quorum sensing pathways are one of several such promising strategies aimed at disarming rather than at eradicating bacterial pathogens during the course of colonization and infection. This review describes mechanisms of bacterial communication involved in biofilm formation. An overview of the potential of marine bacteria and their bioactive components as QS inhibitors is further provided.

## 1. Introduction

Antimicrobial resistance is one of the most serious public health threats that results mostly from the selective pressure exerted by antibiotic use and abuse [1]. The overusage mistakes of the past, notably in the human medicine, veterinary, and agriculture fields, has led to the fast progression and emergence of multi-drug resistance among clinically important bacterial species [2]. Due to this growing increase of resistance, many antimicrobial agents are losing their efficiency. This problem would be of particular concern when bacteria become resistant to all different classes of antibiotics available in the market [3]. Consequently, nowadays therapeutic options for the treatment of infections have become limited or even unavailable, and there are infectious diseases that are almost untreatable by conventional antibiotic therapy [4]. According to a report of the Center for Disease Control and Prevention (CDC), bacterial resistance is still one of the most important causes of morbidity and mortality worldwide [5]. Besides, when first-line and then second-line antibiotic treatment options are limited by resistance, healthcare providers are forced to use upgraded doses and/or combined antibiotic therapy, with inherent toxicity problems for the patient and tremendous socio-economic costs. Even more worrying is the fact that the number of new antibiotics in the drug development pipeline has been continuously declining [6]. Therefore, there is an urgent need to develop strategies that can provide sustainable and long-term effectiveness against resistant pathogenic bacteria [1].

Current therapies rely in general on the ability to kill or inhibit bacterial growth, imposing thus a strong selective pressure on bacteria that can lead to the development of resistance mechanisms [7]. Newer lines of attack should target bacterial cellular processes that are responsible for pathogenesis and virulence instead of components that are essential for growth, which have garnered the name “antipathogenic” or “antivirulence” therapies. Unlike current antibiotics, drugs that target pathogenicity and virulence traits could constitute an advantage for the host immune system regarding bacterial adaptability and infection control. Besides, this approach can provide new means to reduce or avoid the described evolutionary pressure [8,9]. In this way, several strategies have been investigated and proposed based essentially on two master bacterial virulence systems—communication and oxidative weapons assembly systems—which coordinate the whole arsenal of virulence factors [7]. Among them, cell-to-cell communication in bacteria is crucial for their adaptation to different environments and is regulated by quorum sensing (QS) networks [7]. This signalling process is also involved in the expression of genes important to the production of virulence factors, host colonization, biofilm formation and antibiotic resistance in a number of pathogenic bacteria [10]. QS systems play a central role in the ability of bacteria to promote pathogenicity and much attention on the development of new anti-infective agents has been focused on targeting these pathways [7,11]. Therefore, there is a growing interest in finding ways to disrupt, block or manipulate QS signalling in bacteria [12,13]. The interruption of bacterial QS using other organisms, including bacteria, can offer new opportunities to understand better the molecular mechanisms that are behind communication and find alternative therapeutic strategies [13]. This ability to interfere with intercellular communication is a frequent phenomenon in the aquatic environment and is a phenomenon already identified in many marine bacteria, which has attracted the attention of the scientific community.

This study reviews bacterial QS systems and signal molecules with special emphasis on marine pathways. Considering the outstanding biological potential of marine microbial species and the lack of knowledge and exploration regarding the marine environment, the use of marine bacteria and/or their metabolites as QS inhibitors (QSIs) or antagonist is highlighted. A revised discussion of the anti-QS proficiency of marine bacteria and their metabolites will be also presented.

## 2. Quorum Sensing

QS is a process of intercellular communication, being one of the best studied types of interactions among bacterial communities in a diversity of ecological niches (e.g., terrestrial and aquatic) [14]. QS allows bacteria to cooperate or compete with each other (within a species and between species) by coordinating the expression of phenotypes and regulating physiological activities [15,16,17]. They include the production of secondary metabolites, toxins, antibiotics, bioluminescence, extracellular hydrolytic enzymes and exopolysaccharides (essential for bacterial adhesion and biofilm development); sporulation; bacterial conjugation; symbiosis; secretion of virulence factors; biofilm formation/differentiation; and other biological behaviors [16,18,19]. All of these phenotypes are useful for the colonization of different environments or hosts, establishment of disease, acquisition of nutrients and group defense [20]. QS-based interactions are dependent on the cellular density and occur through the production (signal synthase) and sensing (signal receptor) of extracellular chemical signals named autoinducers (AIs) [19]. During bacterial growth, these signaling molecules are continuously produced and released into the surrounding environment until reaching a threshold concentration, also known as “quorum level” [14]. The AIs, are then recognized by specific receptor proteins localized in the cytoplasm (Gram-negative bacteria) or in the membrane (Gram-positive bacteria), triggering a cascade of events that start the transcription of QS-regulated genes [21,22].

Different types of QS signals have been identified (Figure 1) [23,24,25,26,27]. Based on their structure and specific functions they are classified in three classes: (1) the acyl homoserine lactones (AHLs—AI-1)—are small molecules with a lactone ring and acyl side chain, primarily involved in QS mediation by Gram-negative bacteria [28]; (2) the autoinducer peptides (AIPs)—are short peptide chains produced in the cell, requiring membrane transport proteins to cross the cellular membrane and regulate QS in Gram-positive bacteria [29,30]; and autoinducer-2 (AI-2)—are furanone derived signal molecules (e.g., furanosyl borate diester) with combined characteristics of AHLs and AIPs, mediating QS in both Gram-negative and –positive bacteria (“universal” signaling molecules used for interspecies communication) [31].

Additionally, other signaling molecules have been described [32,33] and new chemical signals continue to be discovered. Fatty acids (e.g., diffusible signal factors-DSFs; cis-2-dodecenoic acid) [34,35], *Pseudomonas* quinolone signal (PQS; 2-heptyl-3-hydroxy-4-quinolone) [36,37], partial ester compounds and autoinducer-3 (produced by enterohemorrhagic *Escherichia coli* O157:H7—involved in inter-kingdom communication and host-microbe interactions) are typical examples of these signaling molecules.

The main QS systems described are the LuxI/R-type system, the Agr system and the LuxS/AI-2 system [38]. The LuxI/R-type system utilizes AHLs as chemical signals and thus is mostly used by Gram-negative bacteria. This QS system consist of two components, the LuxI and LuxR proteins that is the AHL synthase and AHL receptor, respectively [18,19]. AHL-mediated QS was firstly discovered in the Gram-negative marine bacterium *Vibrio fischeri*, in which the regulation of the luminescence production is cell-density-dependent [21]. The Agr system is the linguistic communication commonly present in Gram-positive bacteria and uses peptide substances (AIPs) as the signal. It is a two-component QS system (RNA II and RNA III), found for example in *Staphylococcus aureus* [18]. LuxS/AI-2 system is involved in the synthesis of AI-2 and mediate interspecies and intraspecies interactions between Gram-positive and -negative bacteria [39]. The LuxS protein is a homodimeric metallo-enzyme that contains two identical tetrahedral metal-binding sites and can be encountered in *Streptococcus* genus (e.g., *Streptococcus mutans*, *Streptococcus pyogenes*, *Streptococcus pneumonia* and *Streptococcus suis*), *Lactococcus lactis*, *Clostridium perfringens*, *Neisseria meningitidis*, *Escherichia coli*, and *Haemophilus influenza* [39].

Some bacteria have the ability to produce and detect several AIs simultaneously. One classical example is *Vibrio harveyi*. This complex system was first discovered in this marine bacterium, being used as model [40]. Indeed, in *V. harveyi* QS network uses three AIs, depending if it is for intra-species, intra-genera or inter-species communication [19]. Another common canonical network architecture is that found in *Pseudomonas* spp., particularly *Pseudomonas aeruginosa*. In this bacterial species, there are four known QS pathways that work independently/dependently, two of them being of the LuxI/LuxR type (LasI/LasR and RhlI/RhlR systems), the quinolone-based QS system (PQS, 2-heptyl3-hydroxy-4-quinolone signal) and more recently the integrated QS system (IQS, 2-(2-hydroxyphenyl)-thiazole-4-carbaldehyde signal). These QS circuits are hierarchically arranged [41].

QS is commonly associated with functions that stimulate pathogenicity and/or virulence, but it can regulate other unrelated behaviors, particularly in the marine environment [14]. QS phenomenon have a huge impact on a variety of marine microbial systems and thus, has been receiving increasing attention from marine biologists and ecologists [14,38].

### Quorum Sensing in Marine Bacteria

Numerous bacteria from different environments produce QS signaling molecules and have many QS-regulated functions [27]. Although it has been extensively demonstrated that the QS communication is very common for interactions among human pathogens, evidence has also been collected regarding the use of such mechanisms by non-pathogens such as marine bacteria [42]. In fact, most of the information on the production of QS signals by marine bacteria are about *Vibrio* spp. (e.g., *V. fischeri*, *Vibrio anguillarum* and *V. harveyi*), while other species are overlooked [42].

Marine bacteria QS signal producers can be found free-living and associated with invertebrates, sponges and diatoms, belonging manly to the α-Proteobacteria and γ-Proteobacteria groups [42]. The genera *Pseudoalteromonas*, *Thalassomonas*, *Pseudomonas*, *Roseobacter*, *Aeromonas* and *Vibrio* are very common AIs producers in the marine habitat [43,44,45,46,47]. As matter of fact, Gram-negative bacteria are the prevailing bacteria in the marine environment [48].

The classes of QS signals in marine bacteria are predominantly of type I (AI-1) [e.g., AHLs (*V. fischeri*) and α-hydroxyketones-AHKs (*V. harveyi*)] and II (AI-2) [e.g., furanosyl-borate diesters (*V. harveyi* and *V. cholera*)]. These signaling molecules are known to regulate the expression of genes responsible for QS-controlled behaviors: production of bioluminescence, antibiotics, virulence factors, enzymes and biofilm development [14,42] (Figure 2). The best studied QS systems in marine microbial environments occur in surface-attached communities (biofilms) and depend on AHL signaling [14]. The main role of AHL-QS in marine microbial communities is related to ecologically and biogeochemically processes as well as to massive bioluminescence episodes associated with algal blooms [14]. The function of the AI-2-QS remains to be understood. However, its involvement in the regulation of the interspecies interactions in complex microbial communities has already been described [14].

## 3. Quorum Sensing Inhibitors

Undoubtedly, QS inhibition strategies, also known as “quorum quenching” (QQ), have a multifaceted value, particularly in the present scenario of rising antibiotic resistance. Such molecules are valuable to restrain or even preclude the impact of bacterial diseases in plants, animals or humans [24]. In addition to their role in infection control, the signaling molecules can also influence other microbiological features, particularly microbe-microbe interaction, host-pathogen interaction, and microbial physiology. Microorganisms can develop signal interference mechanisms to adapt to different environments, and compete for nutrients and ecological niches [49]. In a clinical perspective, the most relevant aspects of this approach are their no-lethality and versatility, as it exerts a more restricted selective pressure on bacterial survival and can act on several molecular targets [24]. Another proposed advantage is that QSIs can also favor the use of low doses of antibiotics, as they usually improve their effectiveness [10].

The interference with QS processes can be diverse, as it depends on the nature (chemical compounds, enzymes), mode of action and targets involved [27]. To disrupt QS phenomenon, three main steps can be targeted according to the QS circuit where quenching occurs and includes the signal synthase, the signal themselves and the signal receptor/transducer [27]. They can be broadly grouped into two groups, the QSIs (non-enzymatic methods) and the QQ enzymes (enzymatic methods). QSIs generally englobe compounds that are able to inactive AI synthases or receptors by competitive binding/structural modification, while QQ enzymes switch off signal transmission by signal degradation [8,18]. The first major QS-disrupting strategy that has been studied is the interference with the detection of the AIs and the second one is the inactivation/degradations of the signal molecules [50]. Halogenated furanones (e.g., (5Z)-4-bromo-5-(bromomethylene)-3-butyl-2(5H)-furanone) are the first group of QSIs encountered and were obtained from red marine algae *Delisea pulchra* [51]. This algae is one of the organisms that has been well studied for the production of QSIs [27]. Production of QQ enzymes that degrade QS signals have been identified in both eukaryotic and prokaryotic organisms [52]. QQ enzymes of eukaryotic origin were reported in mammals such as human [53,54] and porcine [55], other vertebrates [56] and invertebrates [57,58]. The ability to quench QS signal by enzymes is extensively distributed among bacteria. Expression of QQ enzymes by α-proteobacteria, β-proteobacteria, and the γ-proteobacteria, as well as in some Gram-positive species, has been described [59]. The bacterial species with documented QQ enzymatic activity include *Bacillus* sp., *Bacillus thuringiensis*, *Bacillus. cereus*, *Bacillus mycoides*, *Bacillus anthracis*, *Bacillus licheniformis*, *Bacillus amyloliquefaciens*, *Bacillus megaterium*, *Agrobacterium tumefaciens*, *Arthrobacter* sp., *Klebsiella pneumoniae*, *P. aeruginosa*, *Pseudomonas syringae*, *Rastonia* sp., *Acinetobacter baumannii*, *Variovorax paradoxus*, *Rhodococcus erythropolis*, *Mycobacterium tuberculosis*, *Muricauda olearia*, etc. [52,60,61,62,63]. The greatest part of the QQ enzymes are involved in AHL-degradation, which can be classified into three types based on their catalytic mechanism (Figure 3): AHL lactonase/paraoxonase (lactone hydrolysis), AHL acylase (amidohydrolysis) and AHL oxidase/reductase (oxidoreduction) [18,62]. Most of the described QS inhibition strategies have primarily targeted AI-1 and then AI-2. The first one is directed to only address infections by specific single species and the second permits the simultaneous inhibition and modulation of QS pathways in many species [64].

Screenings for QSIs reveal that they can be synthetic or found in nature from terrestrial, marine or freshwater ecosystems. The synthetic compounds can be tailored from exiting chemical libraries or based in a drug design approach (mostly signal mimics and furanone analogues). In nature, QSIs are generated by a wide range of living organisms, such as plants, animals, fungi or bacteria [11,13,27,65]. The majority of the known QSIs were predominantly identified in plants and bacteria. This could be because both plant extracts and bacteria have been more screened for these activities [27,42]. Indeed, prokaryotic byproducts is one of the strategies popularly adopted to interrupt QS mechanism. Numerous reports on bacterial metabolites with QQ activity have been published, including those from marine sources (some examples will be presented later in this review [18,22,66,67,68]. However, information on marine microbial species is limited when compared to their terrestrial counterparts, and many molecules remain to be identified [66].

### QS Inhibitors from Marine Bacteria

Most of the available drugs in the market are natural-based formulations and this will continue to be one of the leading trends in the future [69]. Among the natural sources, the marine environment offers a plethora of resources (plants and animals) with pharmacological interest that still remain unexplored [70]. Research about the biotechnological potential of marine organisms (comprising corals, sponges, algae and bacteria) is limited and few marine-derived products are in clinical use. Examples of products include, cytarabine (Cytosar-U^®^, 1969; Depocyt^®^; cancer and leukemia), vidarabine (Vira-A^®^, 1979; antiviral—herpes simplex virus), ziconotide (Prialt^®^, 2004; severe chronic pain), omega-3-acid ethyl esters (Lovaza^®^, 2004; hypertriglyceridemia), eribulin mesylate (Halaven^®^, 2010; cancer: metastatic breast cancer), brentuximab vedotin (Adcetris^®^, 2011; cancer: anaplastic large T-cell systemic malignant lymphoma, Hodgkin’s disease), trabectedin (Yondelis^®^, 2015; cancer: soft tissue sarcoma and ovarian cancer) and plitidepsin (Aplidin^®^, 2018; cancer: multiple myeloma, leukemia, lymphoma) [71,72,73,74,75]. The first seven are currently Food and Drug Administration (FDA) approved marine-derived drugs, and the last one was approved by the European Agency [70,74,76] (Figure 4).

However, there are many marine-derived compounds in the different stages of preclinical trials as well as in clinical trials (phases I, II and II), directed for diverse illnesses (antibacterial, antiparasitic, antiviral, antimalarial, anti-inflammatory, analgesic, neuroprotective and anticancer) [70,74,76,77]. Actually, marine organisms produce an enormous diversity of bioactive molecules with distinct chemical structures and functional features (from that found in terrestrial habitat) that provides a potential source of novel pharmaceuticals for the treatment of several diseases [70,78,79]. Some examples of new chemical entities with uncommon structures that can be found in marine bacteria are abyssomicins, salinosporamide A, and enediyne-derived cyanosporaside [80]. Abyssomicins are polycyclic polyketide-type antibiotics produced by actinomycetes of the genus *Verrucosispora*, with the ability to interfere with the biosynthetic pathway of the *p*-aminobenzoate/tetrahydrofolate. This is an interesting target since it occurs in diverse microorganisms and not in humans and few inhibitors of folate metabolism have been identified (e.g., sulfonamides and trimethoprim) [81]. Another recognized rare structural scaffold, also obtained from a marine actinomycete named *Salinispora pacifica*, is the enediyne-derived cyanosporaside. The potent DNA damaging activity and unique biosynthetic assembly is characteristic of enediyne compounds [82]. More recently, the metabolite salinosporamide A, isolated from the bacterium *Salinispora tropica*, was used for cancer chemotherapeutic by targeting the β subunit of the 20S proteasome. In fact, proteasome inhibitors are considered one of the most promising treatment options in cancer therapy [83].

The current focus of marine pharmacology regards the discovery of new drug candidates from marine microorganisms [76,84]. Regarding QSIs, some bacteria (e.g., *Bacillus* sp., *Vibrio* sp., *P. aeruginosa*) and other marine organisms (e.g., coral, sponges and algae) have both the ability to respond to QS signaling molecules of partner bacteria or to interfere and block them [12,42,67]. In fact, marine organisms and their associated bacteria are known to produce secondary metabolites with QS inhibitory properties (Figure 5 and Table 1). Strategies based on both QSIs and QQ enzymes seem to be widespread in the marine environment, including in marine bacteria, highlighting the importance of this biological interference for microbial processes in the ocean [14]. In the marine environment, bacteria usually adopt QS inhibition strategies to achieve competitive advantage, at least in surfaces such as biofilms and eukaryotic niches [85].

Kanagasabhapathy et al. [12] performed a screening for QSIs identification from epibiotic bacteria associated with brown algae *Colpomenia sinuosa*, using *Serratia rubidaea* JCM 14263 as an indicator organism. They showed that several of the isolated bacterial strains (12%) were able to inhibit the production of red pigment by *S. rubidaea* JCM 14263 (QS regulated), suggesting its QS inhibitory activity. These isolates belong to the families Bacillaceae (Firmicutes), Pseudomonadaceae (Proteobacteria), Pseudoalteromonadaceae (Proteobacteria) and Vibrionaceae (Proteobacteria). The observed inhibitory effect is associated with the production of QSIs or QSI-like compounds as a mean of host defense and competition with other bacteria [12].

QQ activity of marine cultivable bacteria isolated from different marine samples, including diatom-dominated biofilm loosely, brown seaweed *Fucus vesiculosus* and the sediment of an inland fish culture tank, was investigated by Romero et al. [86]. They found anti-QS activity for some supernatants of the obtained isolates as result of violacein and light inhibition in biosensors *Chromobacterium violaceum* CV026/*C. violaceum* VIR07 and *E. coli* JM109 pSB1075, respectively. These authors also stated that for some of the isolates with positive results, the activity was related to an enzymatic inactivation (presence of acylases/lactonases). Active isolates were identified as belonging to the phylum/class of Alpha- and Gammaproteobacteria, Actinobacteria, Firmicutes, and Bacteroidetes.

Recently, environmental samples collected from the North Atlantic Ocean were screened with the aim of discovering compounds with QS inhibitory action, produced by the marine bacteria isolated from surface waters. The results showed that amongst the hundreds of isolates screened, some of them inhibited QS-mediated violacein production in *C. violaceum* ATCC12472. Identification of the bacterial strain that promoted the most significant reduction, revealed high similarity (100%) with *Rhizobium* sp. The aqueous and organic extracts of *Rhizobium* sp. strain demonstrated ability to disrupt biofilm formation by *P. aeruginosa* PAO1, to downregulate the production of virulence factors (elastase and siderophore) and increase biofilm susceptibility to antibiotic kanamycin. It was also stated that the active components contained on the *Rhizobium* sp. supernatant were AHL analogues, specifically *N*-butyryl homoserine lactone (C4-AHL), suggesting that the effects observed were due to competition with AIs produced by *P. aeruginosa* PAO1 [87].

Teasdale and coworkers [88] reported that the Gram-positive marine bacterium *Halobacillus salinus*, obtained from a sea grass sample collected at the Rhode Island estuar, secrete secondary metabolites able to interfere with QS-regulated phenotypes in Gram-negative species (bioluminescence production by *V. harveyi* BB120) without causing growth inhibition. In the same way, these authors extracted and purified the active metabolites responsible for the observed effect. The active compounds identified were two phenethylamide, named N-(2′-phenylethyl)-isobutyramide (Figure 5a) and 3-methyl-N-(2′-phenylethyl)-butyramide (Figure 5b). These metabolites were further screened for QS inhibitory activity using QS reporter strains (*V. harveyi* BB120, *C. violaceum* CV026 and *C. violaceum* ATCC 12472) and positive outcomes were obtained (violacein and bioluminescence production inhibition). They also found that while considering the structural and molecular sizes similarity of the two phenethylamide and the AHL AIs, these compounds could be AHL structural mimics and compete for receptor binding. This hypothesis was corroborated for 3-methyl-N-(2′-phenylethyl)-butyramide (Figure 5b) using an *E. coli* JB525 sensor strain. Another study was performed in order to found more Gram-positive marine bacteria capable of producing secondary metabolites able to quench QS-controlled behaviors in Gram-negative reporter strains [89]. For this, a panel of 332 Gram-positive isolates obtained from different marine samples, including algae, invertebrates and surface sediments (collected along the Rhode Island coastline), were tested for interference with *V. harveyi* bioluminescence production, a cell signaling-regulated phenotype. They showed that 49 of the bacterial isolates inhibit bioluminescence production in *V. harveyi* without visible effects on its growth. Additionally, around 28 of the generated metabolic extracts interfered with bioluminescence production in *V. harveyi* and some of them (5 extracts) with violacein production in *C. violaceum*. It was also verified that most of the active bacterial isolates pertained to genus *Bacillus* or *Halobacillus* (only two belonged to *Streptomyces* and *Micromonospora* genera), and phenethylamides and a cyclic dipeptide (Figure 5c) are the two types of secondary metabolites responsible for the activities reported. The presence of the lactonase aiiA gene was detected in some of the isolates suggesting that enzymatic degradation of AHL signaling molecules could be related with the observed QS interference [89]. In another study, Nithya et al. [90] reported the production of anti-QS and antibiofilm substances by marine bacterial isolates (identified as *Bacillus pumilus*, *Bacillus indicus*, *Bacillus arsenicus*, *Halobacillus trueperi*, *Ferrimonas balearica*, and *Marinobacter hydrocarbonoclasticus*) collected from different sediment samples of Palk Bay region (India). They demonstrated that bacterial extracts reduced significantly the production of violacein in *C. violaceum* ATCC 12472/CV026 without growth inhibition and induced *P. aeruginosa* PAO1 biofilm dispersion, with disruption of the biofilm architecture. These alterations in *P. aeruginosa* PAO1 biofilms were found to be correlated with reductions of EPS production and the hydrophobicity index. The purification and characterization of the active principle of the most efficient crude extract of *Bacillus* spp. (*B. pumilus*) revealed the presence of phenolic groups and C–H stretches with amine groups.

Metabolites isolated from marine-derived actinomycetes of the genus *Streptomyces*, collected from shallow-water sediments of the Tongyoung Bay (Korea), were considered mimics of AHL signals and classified as QS signal competitors. These metabolites contain a common lactone moiety combined with a blastmycinolactol, being included in the butenolides (Figure 5d–f) and 3-hydroxy-γ-butyrolactones (Figure 5g–i) classes [91]. Pathogenic species of *Vibrio* spp. such as *V. harveyi*, pose serious problems in aquaculture that lead to enormous loses and thus economic implications for the producers. The problem becomes more critical when pathogenic *Vibrio* strains form biofilms [92]. In this sense, 88 marine actinomycetes isolated from marine sediments of South China were examined for their potential to preclude biofilms formation or to eradicate already stablished biofilms as well as their ability to inhibit QS in *Vibrio* species (*Vibrio vulnificus* V0105, *V. anguillarum* AN0306, and *V. harveyi* H). The authors attested that some extracts inhibited biofilm formation, dispersed mature biofilms or inhibited the QS system of *V. harveyi*. They also found that one of the isolates, identified as *Steptomyces albus*, demonstrated both the ability of attenuate biofilm formation and the activity of AIs AHLs (using *A. tumefaciens*WCF47 (pCF372/pCF218) indicator organism) [92]. Interesting results were also achieved with methanolic extracts of a coral associated actinomycete, identified as *Streptomyces akiyoshiensis*, against *S. aureus* reference strains (including MRSA) and clinical isolates, regarding antibiofilm and anti-QS activities [93]. The QS inhibitory activity was confirmed through the inhibition of the production of violacein pigment in *C. violaceum* ATCC 12472. Reduction in *S. aureus* biofilm formation were observed not only in vitro but also in vivo, which was assessed using the nematode *Caenorhabditis elegans* as an infection model. It is worth mentioning that no antibacterial activity was found, giving the indication that the antibiofilm effect might be associated with an interference with QS pathways. Naik and colleagues [94] investigated the occurrence of QSIs in marine sponge-derived actinomycetes (using *C. violaceum* CV12472 indicator strain) and tested its inhibitory activities against virulence factors (swarming, biofilm formation, pyocyanin, rhamnolipid and LasA production) that are QS-regulated in *P. aeruginosa* ATCC 27853. Methanolic extracts of some of the marine invertebrate–associated *Streptomyces* isolates demonstrated capability to both inhibit the production of violacein by *C. violaceum* and downregulate the QS-mediated virulence factors in *P. aeruginosa*. The chemical analysis of the active methanolic extracts showed the presence of the constituent’s cinnamic acid (Figure 5j), linear dipeptides proline–glycine (Figure 5k) and N-amido-α-proline (Figure 5l). More recently, the anti-QS potential of marine actinomycetes obtained from samples of seawater was explored. It was found that five of the examined strains inhibited the production of violacein by *C. violaceum* ATCC 12472. The most prominent strain was identified as *Streptomyces parvulus*. Extracts of this bacterium demonstrated also ability to preclude biofilm formation by *P. aeruginosa* PAO1, *S. aureus* 95005, *Micrococcus luteus* 95006 and *Ruegeria* sp. 01008. Metabolites of *S. parvulus* were isolated, purified and characterized revealing to be actinomycin D (Figure 5m) and cyclic (4-hydroxy-Pro-Phe) (Figure 5n). The bioactivity of actinomycin D (Figure 5m) was analyzed and demonstrated a QS inhibition effect and a capability to inhibit prodigiosin production by *Serratia proteamaculans* [95].

In the study of Gutiérrez-Barranquero et al. [96], a collection of bacterial isolates obtained from marine sponges demonstrated QS inhibitory potential on three different biosensor reporter strains: *Serratia marcescens* SP15, *C. violaceum* DSM 30191 and *A. tumefaciens* NTL4. The bacterial isolates that inhibited the QS system of at least one biosensor strain were identified and belong to the class Gammaproteobacteria (*Pseudomonas* sp. strains B98C39, B98SK51b, B98SK53b, B98SK52 and B98SM8, *Pseudoalteromonas* sp strains J10, JC29, W3, W11 and W21, and *Psychrobacter* sp. strain B98C22), class Alphaproteobacteria (*Paracoccus* sp. JM45) and phylum Firmicutes (*Bacillus* sp. strains AF46, AAF47, AF52, B9853 and CC32 and *Staphylococcus* sp. strain B98C566). The authors verified that the observed QS inhibitory effects were non-enzymatic. Antibiofilm activity against *P. aeruginosa* PA14 and *Bacillus subtilis* CH8a was also found. Besides, *P. aeruginosa* PA14 virulence determinants, such as swarming/swimming motility and pyocyanin production, were suppressed. Marine bacteria (272 in total) isolated from the sponge tissues were screened for their anti-QS potential, using *C. violaceum* 12,472 reporter strain, and some of them revealed strong inhibitory activity (reduction of violacein production) [97]. The most prominent bacterial extracts were secondary scrutinized and permitted the selection of the highly active bacterial strain that was identified as *Staphylococcus saprophyticus* [98]. After extraction, isolation and identification of *S. saprophyticus* active secondary metabolites, the Cyclo(Pro-Leu) (Figure 5o) showed a moderate QS inhibition effect.

In another work, extracts of 14 different species of sponges collected from the Red (Eilat, Israel) and Mediterranean (Achziv nature marine reserve) seas were screened for their anti-QS activity using two bioreporter strains (*V. fischeri* based selector system and *C. violaceum* CV026), and eleven demonstrated positive outcomes at least for one of strains. Bacterial isolates (phylum Proteobacteria, Firmicutes, Actinobateria and Bacteroidete) of six of the active sponge species were also evaluated for the QS inhibitory effects (*C. violaceum* CV026 and *A. tumefaciens* NT1) and approximately 20% revealed capability to inhibit the QS regulated violacein production. They also found that the most promising bacterial isolates interfered with *P. aeruginosa* PAO controlled virulence factors (pyocyanin and protease production) and with biofilm formation. The metabolic profile of these isolates were also identified and the presence of the compounds licochalcone A (Figure 5p), malyngamide-J (Figure 5q), malyngamide-L (Figure 5r), isomitomycin A (Figure 5s), ansamitocin P-3 (Figure 5t), pederin (Figure 5u), nisamycin (Figure 5v) and kanglemycin A (Figure 5w) were detected. Thus, for some of the sponges the QS inhibitory activity could be related to the presence of bacterial isolates/metabolites [99]. Costantino et al. [100] studied the QS inhibitory activity of a hopanoid derivative named plakohopanoid, which was obtained from the extract of the sponge *Plakortis* cf. *lita*. The identified γ-lactone was probably secreted by bacterial symbionts of this sponge. Using the bacterial bioreporter *E. coli* pSB1075 (long-chain AHL-QS systems), the authors verified that plakohopanoid (Figure 5x) has high potential to inhibit QS controlled bioluminescence only at submicromolar concentrations. Moreover, this metabolite also showed ability to interfere with the virulence of the wild-type *P. aeruginosa* PAO1, as demonstrated by a decrease of the proteolytic activity [100].

Recently, interesting QS inhibition outcomes were achieved with the marine cyclic dipeptide -cyclo(L-leucyl-l-prolyl) (Figure 5o), secreted by a mangrove rhizosphere bacterium (*B. amyloliquefaciens*), against both reference and clinical isolates of *S. marcescens*, based on the inhibition of the prodigiosin production that is QS-controlled [101]. This compound exhibited an extraordinary ability to inhibit the production of other QS-regulated virulence factors (biofilm, exopolymeric substance, protease and lipase) and to affect the hydrophobicity and motility of *S. marcescens*. Additionally, using the *C. elegans* infection model the in vivo anti-adherence capability of the cyclic dipeptide -cyclo(L-leucyl-l-prolyl) (Figure 5o), as well also its ability to down regulate QS controlled virulence genes was verified. Other studies demonstrated the clinical value of mangrove rhizosphere bacteria, which include antibiofilm and antivirulence properties against important foodborne and oral pathogen (e.g., *Listeria monocytogenes*, *S. mutans*) [105,106].

In order to search for new QSI, the coral *Pocillopora damicornis* associated bacterial isolates were examined using the indicator strain *C. violaceum* ATCC 12472 [66]. Positive anti-QS results were obtained for some bacterial extracts and one of them identified as *Staphylococcus hominis*. This extract also demonstrated anti-biofilm activity against clinical isolates of *P. aeruginosa* PAO1 and ability to down-regulate QS regulatory genes. From *S. hominis* extracts a compound with strong QS inhibitory activity, the DL-homocysteine thiolactone that is an analog of the AHLs, was isolated. It was hypothesized that DL-homocysteine thiolactone (Figure 5y) compete with AHL for the binding site [66].

QSIs also have been found in cyanobacteria [102,103,104]. For example, the cyanobacterium *Blennothrix cantharidosmum* produce tumonoic acids (E, F, G and H) (Figure 5z–cc), which are capable to inhibit bioluminescence production in *V. harveyi*, a phenomenon QS dependent [102]. An investigation of Dobretsov et al. [103], carried out with several extracts of marine cyanobacteria collected from Florida waters (marine and estuarine locations), demonstrated that most of them had high ability to disrupt QS as suggested by inhibition of violacein pigment production of the reporter strain *C. violaceum* CV017. The strongest QS inhibitory potential was obtained with extracts of *Symploca hydnoides* and *Lyngbya majuscule*, being the compound malyngolide (Figure 5dd) associated to the inhibitory activity of the last one. The possible QS inhibitory mechanism of the identified QSI malyngolide is the competition for the AHL binding site of the LasR receptor. In another study, Romero et al. [104] showed that filamentous nitrogen-fixing cyanobacterium *Anabaena* (Nostoc) sp. PCC 7120 produced an enzyme of the acylase type that has AHL degradation activity (AHL-acylase). Together, these results give the indication that marine cyanobacteria can be an interesting source of QSIs, requiring further research [42]. In fact, the mode of action of their entire extracts or individual active compounds is not yet properly characterized [103].

## 4. Significance and Future Perspectives

QS plays a central role in the regulation of bacterial functions, including the expression of virulence genes in pathogenic bacteria of importance to clinical, aquaculture and agriculture sectors. Therefore, interfering with the timing of these regulatory pathways can offers new ways (different from these of antibiotics) to develop future strategies for controlling infectious diseases [42]. The manipulation of bacterial QS has attracted considerable interest from different industries (e.g., pharmaceutical, agricultural and aquaculture), as it provides new opportunities for a wide range of applications, such as therapeutic applications (humans and animals) or environmental purposes [50,61,77].

The fact that most of the studies on QS inhibition has been conducted in vitro and under laboratory conditions using essentially domesticated strains is a limitation. In order to counteract this, field investigations in conditions that mimic “real” infections (in vivo studies) are needed [22,42,61]. The lack of standardized methods for the screening of novel QSIs candidates as well as the limited knowledge on the specificity of the identified QSIs continue to be a drawback. The side effects, including toxicological, of QSIs on non-pathogenic bacteria and on the health of humans/animals, are another important criteria that has been restraining its application [107]. In fact, there are several known anti-QS compounds but they remain at the preclinical stage of the drug development process [108].

In practice, the expected application of QSIs will be alone or combined with antibiotics as potentiators/adjuvants. It has been reported that bacteria are more susceptible to antibiotics when they are combined with QSIs. This is an interesting outcome as the use of higher doses of antibiotics or antibiotics of broad-spectrum can be avoided and thus prevent eventual deleterious effects for the health [10].

Antivirulence/antipathogenic approaches such as QS disruptions are usually considered to be more attractive, as they act on pathways that are not essential for the bacterial cell growth. Unlike antibiotics, these lines of attack minimize the emergence of resistance strains providing a sustainable and long-term effectiveness. However, they need to be applied with prudence to limit the selection of more virulent strains. There is evidence that bacteria can develop several mechanism of resistance to QSIs, such as mutations in QS circuits, efflux pumps that can restrict the availability of QSI, inactivation or even modification of the target [109,110]. The development of resistance to QS inhibition approaches will definitely be dependent on the strategy used [61]. The use of inhibitors with multiple biological activities and non-competitive has been suggested as a priority [109,110]. QQ enzymes have ability to disrupt QS without the need to enter into the bacterial cells, whereby the development of resistance can be less probable [61]. Besides, the enzymes act in an independent way regarding signaling receptor (don’t need to bind to a target protein), are usually considered non-toxic and can be incorporated into various matrices without being released [61]. While QS disruption affects bacterial fitness and could induce some selective pressure, affecting certain resistant bacteria, it is usually more moderate than antibiotics and spreads slowly [111].

Overall, the inhibition of QS pathways is undoubtedly promising for combatting multidrug-resistant bacteria. Future directions in this field, regarding the applicability, methods of treatment and delivery, specificity, safety, and costs need be investigated. Considering that marine bacteria and derivative compounds demonstrated high potential as QSIs, they need further attention in order to increase interest on using marine resources for advanced biofilm control.

## Figures and Tables

**Figure 1 marinedrugs-17-00427-f001:**
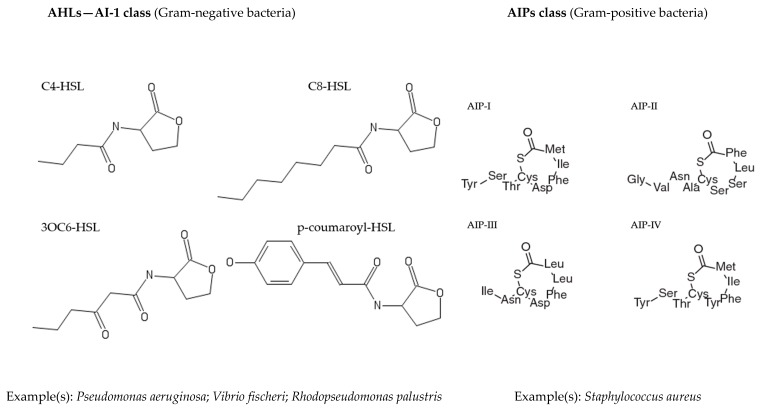
Chemical structures of the main representative types of QS signal molecules used in the microbial regulation of QS. **Note:** C4-HSL—*N*-Butyryl-l-homoserine lactone; C8-HSL—*N*-Octanoyl-l-homoserine lactone; 3OC6-HSL—*N*-(3-Oxohexanoyl)homoserine lactone; p-coumaroyl-HSL—*N*-(4-coumaroyl)-l-homoserine lactone; Non-boron containing AI-2 (R-THMF—*(2R,4S)*-2-methyl-2,3,3,4-tetrahydroxytetrahydrofuran); Boron containing AI-2 (S-THMF-borate—*(2S,4S)*-2-methyl-2,3,3,4-tetrahydroxytetrahydrofuran).

**Figure 2 marinedrugs-17-00427-f002:**
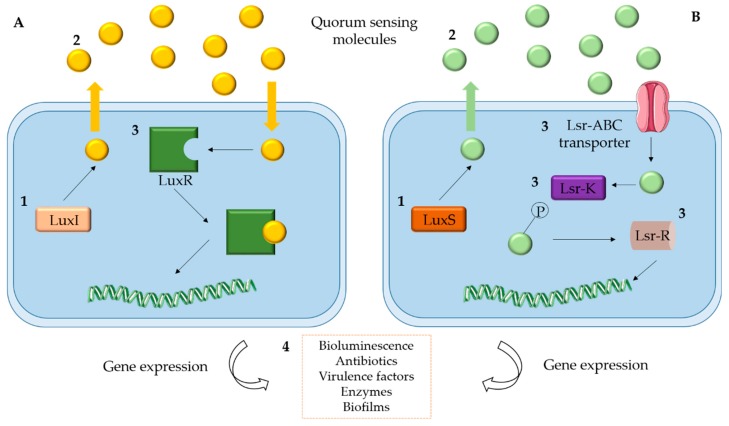
General scheme of the main QS mechanisms described for marine bacteria. (**A**) LuxI/R-type system; (**B**) LuxS/AI-2 system. 1—Signal synthase protein (LuxI, LuxS); 2—Autoinducers (AI-1, AI-2); 3—Response regulator protein/receptor (LuxR; Lsr-ABCKR); 4—QS regulated behaviors. Adapted from Raffa et al. [11].

**Figure 3 marinedrugs-17-00427-f003:**
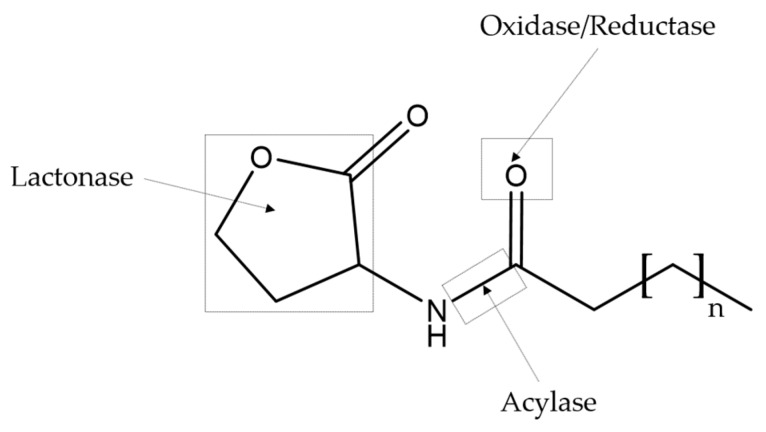
Schematic representation of the AHL-degrading enzyme targets. Broken lines mark position of possible cleavages of *N*-Butyryl-l-homoserine lactone (C4-HSL) molecule by lactonase, acylase and oxidase/reductase.

**Figure 4 marinedrugs-17-00427-f004:**
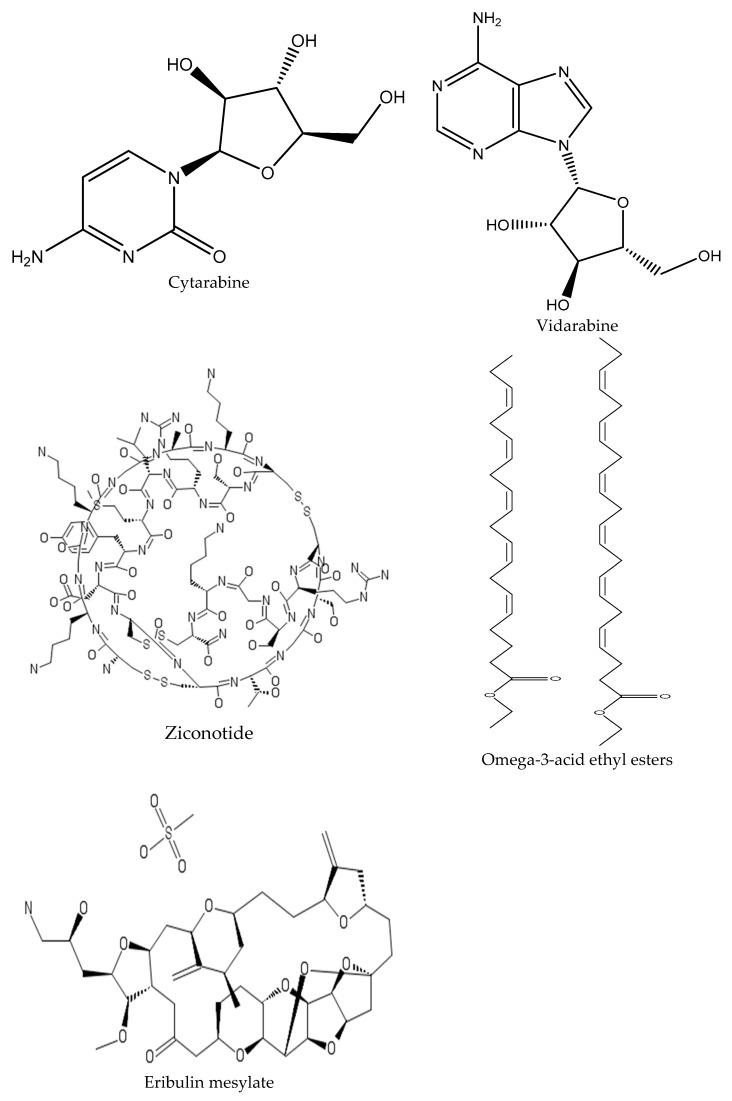
Chemical structures of eight approved marine-derived drugs: cytarabine, vidarabine ziconotide, omega-3-acid ethyl esters, eribulin mesylate, brentuximab vedotin, trabectedin and plitidepsin.

**Figure 5 marinedrugs-17-00427-f005:**
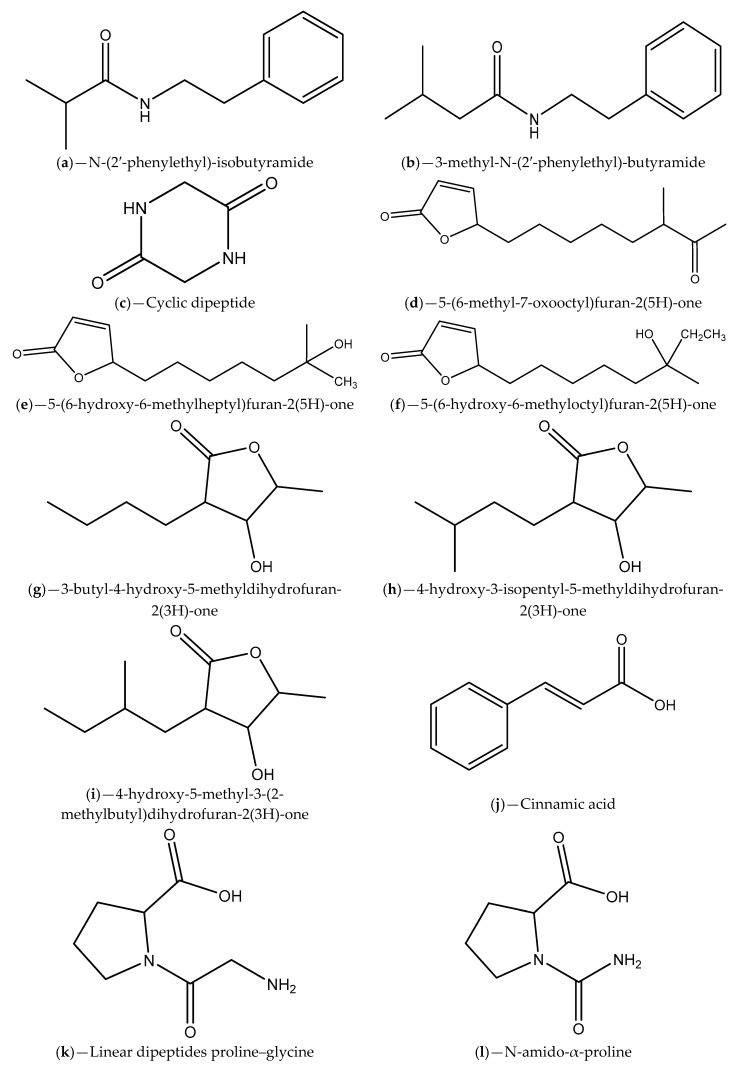
Chemical structures of secondary metabolites associated with bacterial QS inhibitory activities.

**Table 1 marinedrugs-17-00427-t001:** QS inhibition by marine bacteria and their metabolites.

Marine Source	Bacteria and Metabolite(s)	Indicator Organism(s)	QS Inhibitory Activity and QS Related Phenotypes	Reference(s)
Brown algae *Colpomenia sinuosa*	Bacillaceae, Pseudomonadaceae, Pseudoalteromonadaceae, Vibrionaceae families	*S. rubidaea*	Inhibition of red pigment that can be associated with the production of QSIs or QS like compounds	[12]
Diatom-dominated biofilm loosely, brown seaweed *Fucus vesiculosus* and the sediment of an inland fish culture tank	Alpha- and Gammaproteobacteria classes; Actinobacteria, Firmicutes and Bacteroidetes phylum	*C. violaceum* (CV026, VIR07) and *E. coli* (JM109 pSB1075	Inhibition of violacein production and light emission, which can be due to enzymatic inactivation	[86]
Surface water samples collected from the North Atlantic Ocean	*Rhizobium* sp; AHL analogues (C4-AHL)	*C. violaceum* ATCC12472	Inhibition of violacein production. Interference with *P. aeruginosa* biofilm formation, downregulation of virulence factors production and enhanced biofilm susceptibility to antibiotics. The observed outcomes are attributed to AIs competition	[87]
Sea grass sample collected from a Rhode Island estuary	*H. salinus*; N-(2′-phenylethyl)-isobutyramide and 3-methyl-N-(2′-phenylethyl)-butyramide	*V. harveyi* BB120; *V. harveyi* BB120, *C. violaceum* CV026 and *C. violaceum* ATCC 12472	Inhibition of violacein production and luminescence emission, which can be due to receptor binding competition	[88]
Marine samples (algae, invertebrates and surface sediments) collected from Rhode Island coastline	*Bacillus*, *Halobacillus*, *Streptomyces* and *Micromonospora* genera; phenethylamides and a cyclic dipeptide	*V. harveyi*; *C. violaceum*	Inhibition of bioluminescence emission and violacein production that can be related to AHL degradation	[89]
Sediment samples from Palk Bay region	*B. pumilus*, *B. indicus*, *B. arsenicus*, *H. trueperi*, *F. balearica*, and *M. hydrocarbonoclasticus*	*C. violaceum* ATCC 12472/CV026	Inhibition of violacein production; Dispersion of *P. aeruginosa* biofilms	[90]
Shallow-water sediments from Tongyoung Bay	*Streptomyces* genus; Butenolides and 3-hydroxy-γ-butyrolactones	-	Competition with AHL signaling molecules	[91]
Marine sediments from South China	Actinomycetes (e.g., *S. albus*)	*A. tumefaciens*WCF47 (pCF372/pCF218)	Inhibition of biofilm formation and biofilm dispersion in *V. vulnificus* V0105, *V. anguillarum* AN0306, and *V. harveyi* H; Decrease of AHLs activity	[92]
Coral associated actinomycete	Actinomycetes (e.g., *S. akiyoshiensis*)	*C. violaceum* ATCC 12472	Inhibition of violacein production; Inhibition of *S. aureus* (reference strains and clinical isolates) biofilm formation	[93]
Marine sponge-derived actinomycetes	*Streptomyces*; Cinnamic acid, linear dipeptides proline–glycine and N-amido-α-proline	*C. violaceum* CV12472	Inhibition of violacein production; Inference with QS-regulated virulence factors in *P. aeruginosa ATCC 27853*	[94]
Seawater samples from Lianyungang region	Actinomycetes (e.g., *S. parvulus*); Actinomycin D and cyclic (4-hydroxy-Pro-Phe)	*C. violaceum* ATCC 12472; *S. proteamaculans*	Inhibition of violacein production; Inhibition of *P. aeruginosa PAO1, S. aureus 95005, M. luteus 95006 and Ruegeria sp. 01008* biofilm formation; Inhibition of prodigiosin in *S. proteamaculans*	[95]
Marine sponges	Gammaproteobacteria and Alphaproteobacteria classes; Firmicutes phylum	*S.marcescens* SP15, *C. violaceum* DSM 30191, *A. tumefaciens* NTL4	Decrease of AHLs activity (hort-, medium- and long-chain AHLs); Antibiofilm activity against *P. aeruginosa* PA14 and *B. subtilis* CH8a; Inhibition of swarming/swimming motility and pyocyanin production	[96]
Sponge tissues	*S. saprophyticus*; cyclo(Pro-Leu)	*C. violaceum* 12472	Inhibition of violacein production	[97,98]
Sponge tissues	Proteobacteria, Firmicutes, Actinobateria and Bacteroidete phylum; Licochalcone A, malyngamide-J, malyngamide-J, isomitomycin A, ansamitocin P-3, pederin, nisamycin and kanglemycin A	*V. fischeri* based selector system and *C. violaceum* CV026; *A. tumefaciens* NT1	Inhibition of bioluminescence/violacein production; Interference with *P. aeruginosa* QS-regulated virulence factors (pyocyanin and protease production); Inhibition of *P. aeruginosa* biofilm formation	[99]
Marine sponge *Plakortis* cf. *lita*	Bacterial symbionts; plakohopanoid	*E. coli* pSB1075	Inhibition of bioluminescence production; Interference with *P. aeruginosa* QS-regulated virulence factor (total protease activity)	[100]
Mangrove rhizosphere	*B. amyloliquefaciens*; cyclo(L-leucyl-l-prolyl)	*S. marcescens*	Inhibition of prodigiosin in *S. marcescens*; Interference with *S. marcescens* QS-regulated virulence factors	[101]
Coral *Pocillopora damicornis*	*S. hominis*; DL-homocysteine thiolactone	*C. violaceum* ATCC 12472	Inhibition of violacein production; Anti-biofilm activity against *P. aeruginosa* PAO1; Downregulation of *P. aeruginosa* QS-related genes; QSI can be related with AHL competition for the receptor	[66]
Extracts Cyanobacterium *Blennothrix cantharidosmum*	Tumonoic acids (E, F, G and H)	*V. harveyi*	Inhibition of bioluminescence production	[102]
Florida waters samples	Marine cyanobacteria (e.g., *S. hydnoides* and *L. majuscule*); malyngolide	*C. violaceum CV017*	Inhibition of violacein production that can be associated with competition for the binding site	[103]
Filamentous nitrogen-fixing cyanobacterium	*Anabaena* (Nostoc) sp. PCC 7120	-	Degradation of the AI AHL by acylase enzyme	[104]

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
