# Peer review of "Quorum Sensing Inhibition by Marine Bacteria"

_marinedrugs, 2019, doi:10.3390/md17070427_

Reviewer 1 Report

The authors present a nice review on quorum sensing inhibition by bioactive compounds from marine organisms.

I made just few comments that will be usefull to increase the comprehension and the fluidity of the text.

line 138: Which marine bacteria produce AI-1 or AI-2 signal molecules? I think that some examples should be mentioned also for the comprehension of the Figure 1. 

line 151: I think that this section should be number as 3. 

line 201: This section should be called "QS inhibitors from marine organisms"  and numbered 3.1 and to allow a better comprehension and to increase the fluidity of the text the authors could add others sub-sections dividing the QS inhibitors by sources for example. 

line 273: gene name in italic 

line 374: section 4

Author Response

- The authors made a careful revision of the manuscript taking into account the suggestions and comments proposed. All the required modifications were performed. We consider that the comments were very useful to improve the quality of the work.

We made the revision using the following approach:

Our answers to the comments and recommendations proposed by the referee;

Reformulation of the manuscript when needed (with modifications highlighted in blue).

Reviewers' comments to author:

Reviewer 1:

The authors present a nice review on quorum sensing inhibition by bioactive compounds from marine organisms. I made just few comments that will be useful to increase the comprehension and the fluidity of the text.

- The authors acknowledge the Reviewer for the encouraging comments.

Line 138: Which marine bacteria produce AI-1 or AI-2 signal molecules? I think that some examples should be mentioned also for the comprehension of the Figure 1.

- The authors acknowledge the Reviewer comments. This remark was taken into account in the revised manuscript and some examples of marine bacteria were added.

Line 151: I think that this section should be number as 3.

- The authors acknowledge the Reviewer comments. This suggestion was taken into account in the revised manuscript.

Line 201: This section should be called "QS inhibitors from marine organisms" and numbered 3.1 and to allow a better comprehension and to increase the fluidity of the text the authors could add others sub-sections dividing the QS inhibitors by sources for example.

- The authors understand the Reviewer comment. However, in our opinion the amount of available studies does not justifies sub-section division. In fact, such division and information on the source is provided in the table.

Line 273: gene name in italic.

- The authors acknowledge the Reviewer comments. The correction was made.

Line 374: section 4.

- The authors acknowledge the Reviewer comments. This suggestion was taken into account in the revised manuscript.

Reviewer 2 Report

The present manuscript is devoted to a review on the topic of “Quorum sensing inhibition by marine bacteria”. 
The theme of the review is interesting, important, and relevant. There are several published reviews on this topic (for example, doi:10.3390/md17050275; https://doi.org/

10.1128/mBio.02331-17; doi:10.3390/md17020080; doi:10.3390/md15030053 ; http://dx.doi.org/10.1080/08927014.2011.609616 ). 
What is the novelty of this review? 

First of all, this review is based mostly on many others already published reviews than on original research papers.
The presented review consists of three parts. It includes an “Introduction” (part 1), a “Quorum Sensing” (part 2), and the “Significance and future perspectives” (part 3). 
There are one Figure and one Table in the manuscript. The Figure illustrates a “General scheme of the main QS mechanisms described for marine bacteria”. The Table provides information about QS inhibition by marine bacteria and their metabolites. The table is quite informative. 
There are some concerns about this review and several recommendations.

Reader would like to find more information that is specific in the text of the review. However, authors just mention some data without any discussion and without sharing of detail information.

For example:

1)      Page 2. Lines 86-95

Authors mention different types of QS signal molecules in the text. It will be useful to give the concrete examples and to show their structures and provide corresponding references to literature.

2)      Page 4. Lines 175-176. “ …Production of QQ enzymes that degrade QS signal has been identified in both eukaryotes and prokaryotes organisms”.  It should be written:”… in both eukaryotic and prokaryotic organisms”. Moreover, it will be helpful to give the examples of such organisms and provide corresponding references.

3)      Page 5. Lines 184-189. Please, provide a scheme of three types QQ enzymes that are involved in AHL-degradation.

Lines 197-198. Please, provide the concrete information about bacterial metabolites in the text or in the separate Table.

4)      In the Section 2.1.1. “Marine Bacteria” authors discuss the results of several experimental studies. This section is the most interesting and valuable in this review.

      However, some concrete information is missing. So, on line 206 a few marine-derived “products are in clinical use” have been mentioned. It will be good to know: which ones? Please, name them and provide a reference.

Lines 207-212. It will be useful to make a Table and to show in it what kind of bioactive molecules and what chemical structures and features they have.

The same comments could be referred to the text on lines 214-216. Please, provide concrete examples of such bacteria and other marine organisms.

 5) Information about the Secondary metabolites (lines 341, 342, 357,360, 366) could be presented in a Figure or in a Table. It has to be pointed what the source is, what the target is, and

the corresponding references have to be provided.

6) Lines 401-402. “There are evidences that bacterial develop several mechanisms of resistance to QIs”  Please, indicate which ones?

This review is good as an introduction into the problem of a quorum sensing inhibition by marine bacteria. To make this review more solid and more informative it would be useful not only very often to make references to other published reviews, but to provide more concrete details for readers, for example,  to show the chemical formulas of various substances involved in the suppression of QS. It would be helpful to present some Figures or Tables illustrated these compounds. 

Author Response

- The authors made a careful revision of the manuscript taking into account the suggestions and comments proposed. All the required modifications were performed. We consider that the comments were very useful to improve the quality of the work.

We made the revision using the following approach:

Our answers to the comments and recommendations proposed by the referee;

Reformulation of the manuscript when needed (with modifications highlighted in blue).

Reviewers' comments to author:

Reviewer 2:

The present manuscript is devoted to a review on the topic of “Quorum sensing inhibition by marine bacteria”. The theme of the review is interesting, important, and relevant. There are several published reviews on this topic (for example: doi:10.3390/md17050275; https://doi.org/10.1128/mBio.02331-17; doi:10.3390/md17020080;

doi:10.3390/md15030053; http://dx.doi.org/10.1080/08927014.2011.609616).

What is the novelty of this review? First of all, this review is based mostly on many others already published reviews than on original research papers.

The presented review consists of three parts. It includes an “Introduction” (part 1), a “Quorum Sensing” (part 2), and the “Significance and future perspectives” (part 3). There are one Figure and one Table in the manuscript. The Figure illustrates a “General scheme of the main QS mechanisms described for marine bacteria”. The Table provides information about QS inhibition by marine bacteria and their metabolites. The table is quite informative.

- The authors acknowledge the Reviewer comments. Due to the global relevance of the antimicrobial resistance phenomenon and the new possibilities to counteract this problem using antipathogenic/antivirulence therapies based on the interruption of quorum sensing pathways, manuscripts addressing this topic are published in a daily basis. However, the lack of exploration of the marine environment, particularly the potential of marine microorganisms, compared to the terrestrial ones is evident. This is reflected by the reduced number of marine-derived drugs that reached the market and/or are in clinical use (Skindersoe, M. E., Ettinger-Epstein, P., Rasmussen, T. B., Bjarnsholt, T., de Nys, R. & Givskov, M. 2008. Quorum sensing antagonism from marine organisms. Marine Biotechnology, 10(1), pp. 56-63; Malve, H. 2016. Exploring the ocean for new drug developments: marine pharmacology. Journal of Pharmacy & Bioallied Sciences, 8(2), pp. 83-91).  This review is focused on marine quorum sensing, with particular emphasis on the available literature from the use of marine bacteria with quorum sensing inhibitory action. This is significantly different from many of the available reviews whose focus is on the microbial secondary metabolites acting on quorum sensing mechanisms. As recommended, new information was added along the revised manuscript in order to improve the overall scientific quality of this study.

There are some concerns about this review and several recommendations.

Reader would like to find more information that is specific in the text of the review. However, authors just mention some data without any discussion and without sharing of detail information. For example:

1) Page 2. Lines 86-95. Authors mention different types of QS signal molecules in the text. It will be useful to give the concrete examples and to show their structures and provide corresponding references to literature.

- The authors acknowledge the Reviewer comments. The chemical structures of some representative quorum sensing signal molecules was inserted in the revised version of the manuscript.

2) Page 4. Lines 175-176. “…Production of QQ enzymes that degrade QS signal has been identified in both eukaryotes and prokaryotes organisms”. It should be written:”… in both eukaryotic and prokaryotic organisms”. Moreover, it will be helpful to give the examples of such organisms and provide corresponding references.

- The authors acknowledge the Reviewer comments. Additional studies were added in the new version of the manuscript with reference to these aspects.

3) Page 5. Lines 184-189. Please, provide a scheme of three types QQ enzymes that are involved in AHL-degradation. Lines 197-198. Please, provide the concrete information about bacterial metabolites in the text or in the separate Table.

- The authors acknowledge the Reviewer comments. A schematic representation of the types of AHL-enzymes was provided in the revised version of the manuscript. Besides, examples of metabolites and the corresponding chemical structures were provided in the revised manuscript.

4) In the Section 2.1.1. “Marine Bacteria” authors discuss the results of several experimental studies. This section is the most interesting and valuable in this review.

However, some concrete information is missing. So, on line 206 a few marine-derived “products are in clinical use” have been mentioned. It will be good to know: which ones? Please, name them and provide a reference.

- The authors acknowledge the Reviewer comments. In the revised version of the manuscript, examples of marine-derived products are provided as well as the corresponding chemical structures.

Lines 207-212. It will be useful to make a Table and to show in it what kind of bioactive molecules and what chemical structures and features they have.

- The authors acknowledge the Reviewer comments. Some examples of molecules with uncommon characteristics and functions were added in the revised version of the manuscript.

The same comments could be referred to the text on lines 214-216. Please, provide concrete examples of such bacteria and other marine organisms.

- The authors acknowledge the Reviewer comments. Information concerning this comment was added in the revised version of the manuscript.

5) Information about the Secondary metabolites (lines 341, 342, 357,360, 366) could be presented in a Figure or in a Table. It has to be pointed what the source is, what the target is, and the corresponding references have to be provided.

- The authors acknowledge the Reviewer comments. The chemical structure of all mentioned secondary metabolites was included in the revised version of the manuscript.

6) Lines 401-402. “There are evidences that bacterial develop several mechanisms of resistance to QIs” Please, indicate which ones?

- The authors acknowledge the Reviewer comments. Described mechanisms of resistance to QSIs that can occur was added to the revised version of the manuscript.

This review is good as an introduction into the problem of a quorum sensing inhibition by marine bacteria. To make this review more solid and more informative it would be useful not only very often to make references to other published reviews, but to provide more concrete details for readers, for example, to show the chemical formulas of various substances involved in the suppression of QS. It would be helpful to present some Figures or Tables illustrated these compounds.

- The authors acknowledge the Reviewer comments. More figures illustrating the compounds` chemical structures were included in the revised version of the manuscript.